

# Foliar fungal communities strongly differ between habitat patches in a landscape mosaic

Thomas Fort[1], Cécile Robin[1], Xavier Capdevielle[1], Laurent Delière[2] and Corinne Vacher[3]

[1] BIOGECO, UMR 1202, INRA, Université de Bordeaux, Cestas, France
[2] SAVE, UMR 1065, INRA, ISVV, Université de Bordeaux, Villenave d'Ornon, France
[3] BIOGECO, UMR 1202, INRA, Université de Bordeaux, Pessac, France

## ABSTRACT

**Background:** Dispersal events between habitat patches in a landscape mosaic can structure ecological communities and influence the functioning of agrosystems. Here we investigated whether short-distance dispersal events between vineyard and forest patches shape foliar fungal communities. We hypothesized that these communities homogenize between habitats over the course of the growing season, particularly along habitat edges, because of aerial dispersal of spores.

**Methods:** We monitored the richness and composition of foliar and airborne fungal communities over the season, along transects perpendicular to edges between vineyard and forest patches, using Illumina sequencing of the Internal Transcribed Spacer 2 (ITS2) region.

**Results:** In contrast to our expectation, foliar fungal communities in vineyards and forest patches increasingly differentiate over the growing season, even along habitat edges. Moreover, the richness of foliar fungal communities in grapevine drastically decreased over the growing season, in contrast to that of forest trees. The composition of airborne communities did not differ between habitats. The composition of oak foliar fungal communities change between forest edge and centre.

**Discussion:** These results suggest that dispersal events between habitat patches are not major drivers of foliar fungal communities at the landscape scale. Selective pressures exerted in each habitat by the host plant, the microclimate and the agricultural practices play a greater role, and might account for the differentiation of foliar fugal communities between habitats.

Corresponding author
Thomas Fort,
thomaslc.fort@gmail.com

## INTRODUCTION

Plant leaves provide one of the largest microbial habitats on Earth (*Ruinen, 1956*; *Morris, 2001*; *Vorholt, 2012*). They harbour highly diverse microbial communities, including many genera of bacteria and fungi (*Lindow & Leveau, 2002*; *Vorholt, 2012*; *Turner, James & Poole, 2013*). The eco-evolutionary processes which shape these communities—dispersal, evolutionary diversification, selection and drift—are increasingly

well understood (*Hanson et al., 2012*; *Nemergut et al., 2013*; *Vacher et al., 2016*). This new eco-evolutionary framework will undoubtedly have important applications in agriculture. Indeed, crop performance depends on the balance and interactions between pathogenic and beneficial microbial species (*Newton et al., 2010*; *Newton, Gravouil & Fountaine, 2010*). Manipulating whole foliar microbial communities, by acting on the processes shaping them, could thus greatly improve crop health (*Newton et al., 2010*; *Xu et al., 2011*). However, to reach this aim, a better understanding of the structure and dynamics of foliar microbial communities at the landscape scale is required.

The landscape plays a key role in the dynamics of macro-organism populations interacting with crops, such as arthropod pests or their natural enemies (*Norris & Kogan, 2000*; *Chaplin-Kramer et al., 2011*). In ecology, the landscape is defined as an heterogeneous geographic area, characterized by a dynamic mosaic of interacting habitat patches (*Bastian, 2001*). Species movements between habitat patches–referred as dispersal (*Vellend, 2010*)–modulates the richness, composition and function of macro-organism communities (*Hurst et al., 2013*; *Ma et al., 2013*; *Lacasella et al., 2015*). In agricultural landscape, species dispersal between natural and managed habitats can trigger detrimental or beneficial effects in crops (*Chaplin-Kramer et al., 2011*; *Blitzer et al., 2012*), particularly along the edges (*Thomson & Hoffmann, 2009*; *Lacasella et al., 2015*).

The influence of dispersal events on the structure of foliar microbial communities at the landscape scale has hardly been studied. Many microbial species colonising plant leaves are horizontally transferred (i.e. from one adult plant to another) by airborne dispersal (*Whipps et al., 2008*; *Bulgarelli et al., 2013*), while others can come from the seeds, the rhizosphere or the twigs (*Vorholt, 2012*). The foliar microbial communities of a given plant can therefore be influenced by those of its neighbours. Plant pathogens, for instance, can be transmitted from a reservoir plant to neighbouring plants (*Power & Mitchell, 2004*; *Beckstead et al., 2010*; *Wilson et al., 2014*). These short-distance dispersal events could have a greater effect on the foliar microbial communities of annual or deciduous plants, because the leaves of those plants are colonised by micro-organisms every spring, after budbreak.

In this study, we analysed the structure and dynamic of foliar and airborne fungal communities in a heterogeneous landscape consisting of vineyard and forest patches in the south west of France. Vineyards are human-engineered agro-ecosystems, characterized by a low specific and genetic diversity, and where weeds, pests and pathogens are regularly controlled with different cultural practices and pesticides to preserve yield and to reduce infection of leaves and grapes. Conversely, deciduous forests in this area remain little managed and much less homogeneous. We expected the fungal communities of forest patches to be richer than those of vineyards, because the higher plant species richness and biomass in forests increase the diversity of micro-habitats available to foliar fungi. We also expected repeated dispersal events to homogenize foliar fungal communities between the two habitats over the course of the growing season, particularly along habitat edges. We thus tested the following hypotheses for both foliar and airborne fungal communities: (1) community richness is higher in forests than in adjacent

vineyards, (2) community similarity between the two habitats increase over the course of the growing season and (3) is higher along habitat edges.

## MATERIALS AND METHODS

### Sampling design

Three study sites, each consisting of a forest patch and an adjacent vineyard, were selected in the Bordeaux area (France). They were located in the domains of Châteaux Reignac (N44°54′03″, O0°25′01″), Grand-Verdus (N44°47′21″, O0°24′06″) and Couhins (N44°45′04″, O0°33′53″) (Fig. 1A). At each site, the edge between the forest patch and the vineyard was at least 100 m long. The width of each habitat, perpendicular to the edge, was at least 200 m. The forest patches at all three sites contained mostly deciduous species, dominated by pedunculate oak (*Quercus robur* L.). The second most frequent tree species was European hornbeam (*Carpinus betulus* L.) in Reignac and Grand-Verdus, and sweet chestnut (*Castanea sativa* Mill.) in Couhins. In the vineyards, the grapevine (*Vitis vinifera* L.) cultivar was Cabernet Sauvignon in Reignac and Grand-Verdus, and Merlot in Couhins.

At each site, leaves were collected along three parallel transects perpendicular to the forest-vineyard edge and separated by a distance of about five meters (Fig. 1B). Leaves were sampled at four locations along each transect: in the centre of the forest (100 m away from the edge), at the edge of the forest, at the edge of the vineyard and in the centre of the vineyard (100 m away from the edge). In forest patches, leaves were sampled from the two most abundant tree species. For each sampling location and each transect, a single tree of each species was selected. Three leaves oriented in different directions were collected from each tree, at a height of 7 m. In vineyards, three leaves were collected from three adjacent cloned grapevine stocks. Each of the sampled leaves was selected from the base of the cane (one-year-old shoot), to ensure the collection of leaves of the same age on each date. The leaves were removed with scissors that had been sterilised with 96% ethanol, and all contact of the leaves with the hands was carefully avoided. The leaves were stored in clear plastic bags containing silica gel to ensure rapid drying. In addition, grapevine leaves were placed between two sheets of sterile paper filter to ensure good desiccation despite their thickness. Leaves were sampled on three dates in 2013: in May (between the 15th and 23rd), July (between the 16th and 18th) and October (3rd). The sampling dates chosen were as far removed as possible from the last chemical treatment performed in the vineyard (Table S1).

Airborne particles were collected along the middle transect of each site, with two Coriolis air sampler devices positioned one meter above the ground. At each sampling location, three successive 10 min sampling sessions were carried out, with a flow rate of 200 l/min.

### DNA extraction and sequencing

Sample contamination was prevented by exposing all tools and materials required for sample processing and DNA extraction to UV light for 30 min in a laminar flow hood. Four discs (each 8.0 mm in diameter) were cut randomly from each leaf, in the flow

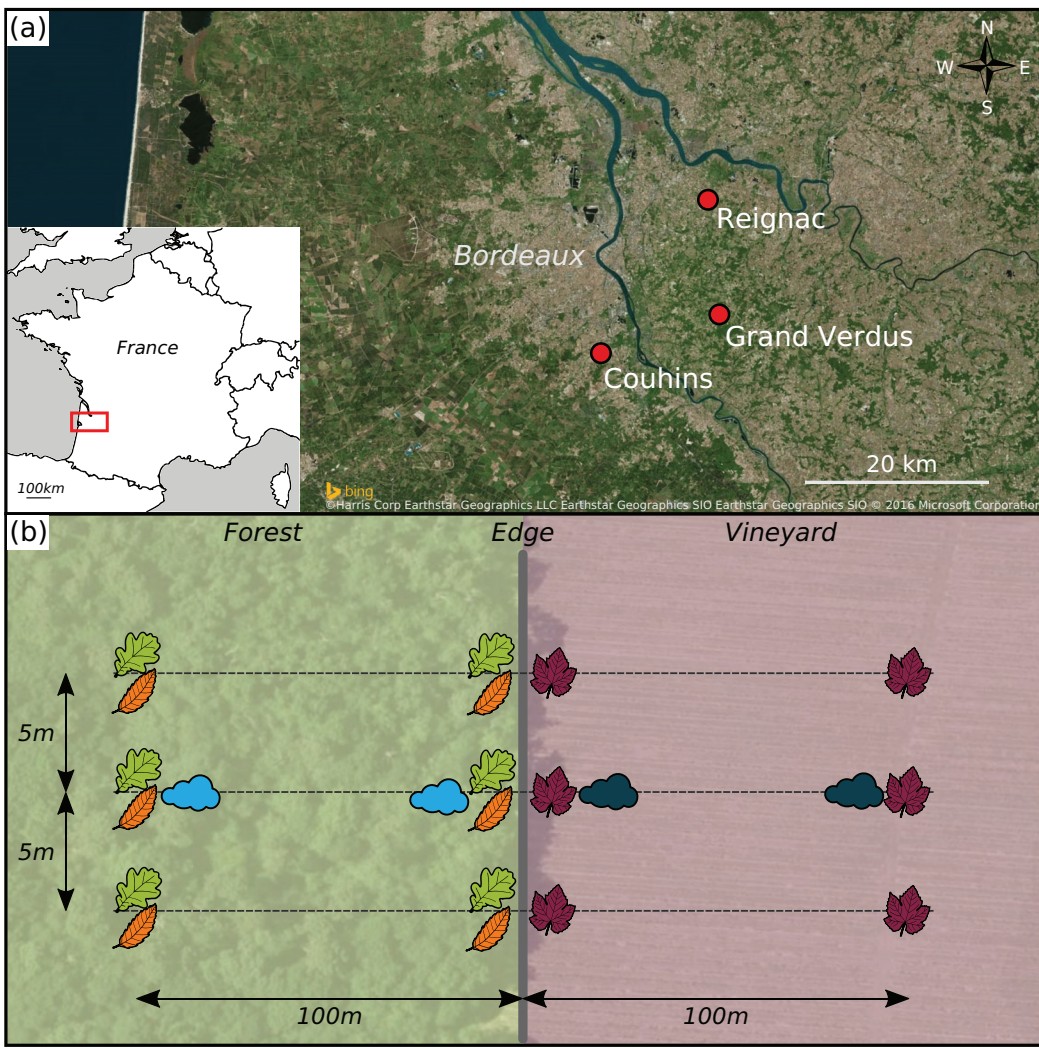

**Figure 1 Experimental design.** (A) Geographical position of the three sampling sites, represented by red points. (B) Sampling design at each site. Leaf pictograms represent the sampling location of leaves in each site. Three leaves per plant species (i.e. grapevine in the vineyard and oak *plus* chestnut or hornbeam in the forest patch) were sampled at each location. Cloud pictograms represent the sampling location of airborne communities.

hood, with a hole-punch sterilised by flaming with 95% ethanol. The four discs were placed in a single well of an autoclaved DNA extraction plate. Three wells were left empty as negative controls. Two autoclaved metallic beads were added to each well, and the plant material was ground into a homogeneous powder with a Geno/Grinder 2010 (SPEX Sample Prep, Metuchen, NJ, USA).

The liquid used to collect airborne particles was transferred into sterile 15 ml centrifuge tubes. Each tube was then centrifuged for 30 min at 13,000 RCF and the supernatant was removed with a sterile transfer pipette. The pellet was then transferred by resuspension to an autoclaved tube and freeze-dried. A tube of unused sampling liquid was treated in the same way and used as a negative control. Total DNA was extracted from each leaf and airborne sample with the DNeasy 96 Plant Kit (QIAGEN).

Foliar DNA samples from the same tree were pooled, as were foliar DNA samples from the three adjacent grapevine stocks.

Fungal Internal Transcribed Spacer 2 (ITS2) was amplified with the fITS7 (forward) and ITS4 (reverse) primers (*Ihrmark et al., 2012*). Paired-end sequencing (300 bp) was then performed in a single run of an Illumina MiSeq sequencer, on the basis of V3 chemistry. Polymerase Chain Reaction (PCR) amplification, barcodes and MiSeq adapters addition, library sequencing and data preprocessing were carried out by the LGC Genomics sequencing service (Berlin, Germany). Sequences were deposited in the European Nucleotide Archive (ENA) database, under the PRJEB13880 project accession number.

## Bioinformatic analysis

Sequences were first demultiplexed and filtered. All sequences with tag mismatches, missing tags, one-sided tags or conflicting tag pairs were discarded. Tags and Illumina TruSeq adapters were then clipped from all sequences, and sequences with a final length fewer than 100 bases were discarded. All sequences with more than three mismatches with the ITS2 primers were discarded. Primers were then clipped and the sequence fragments were placed in a forward-reverse primer orientation. Forward and reverse reads were then combined, and read pair sequences that could not be combined were discarded.

The pipeline developed by *Bálint et al. (2014)* was used to process the sequences. The ITS2 sequence was first extracted from each sequence with the FungalITSextractor (*Nilsson et al., 2010*). All the sequences were then concatenated into a single fasta file, after adding the sample code in the label of each sequence. The sequences were dereplicated, sorted and singletons were discarded with VSEARCH (https://github.com/torognes/vsearch). The sequences were then clustered into molecular operational taxonomic units (OTUs) with the UPARSE algorithm implemented in USEARCH v8 (*Edgar, 2013*), with a minimum identity threshold of 97%. Additional chimera detection was performed against the UNITE database (*Kõljalg et al., 2013*), with the UCHIME algorithm implemented in USEARCH v8 (*Edgar et al., 2011*). The OTU table, giving the number of sequences of each OTU for each sample, was created with USEARCH v8.

OTUs were taxonomically assigned using the online BLAST web interface (*Madden, 2013*) against the GenBank database, by excluding environmental and metagenome sequences. The assignment with the lowest e-value was retained. The full taxonomic lineage of each assignment was retrieved from the GI number information provided by NCBI. All the OTUs assigned to plants or other organisms, and all unassigned OTUs were removed, to ensure that only fungal OTUs were retained.

## Statistical analyses

All statistical analyses were performed in the R environment. We computed 100 random rarefied OTU matrices, using the smallest number of sequences per sample as a threshold. The number of OTUs per sample (OTU richness) and the dissimilarity between samples (Bray-Curtis index based on abundances and Jaccard index based on occurrences) were calculated for each rarefied matrix and averaged (*Cordier et al., 2012*; *Jakuschkin et al., 2016*). However, because the relevance of rarefaction is debated in the scientific

community (*Hughes & Hellmann, 2005*; *McMurdie & Holmes, 2014*), we also performed the analyses on the raw OTU matrix by including the square root of the total number of sequences per sample (abundance) as first explanatory variable in all the models.

Type III ANOVA, which tests for the presence of an effect, given the other effects and the interactions (*Herr, 1986*), was used to assess the effect of host plant species (grapevine, oak, hornbeam and chestnut), sampling date (May, July, October), edge (habitat centre or edge) and their interactions on foliar OTU richness. Sampling site was included in the model as a random factor. Marginal and conditional coefficients of determination were calculated to estimate the variance explained by fixed factors ($R_m^2$) and fixed *plus* random factors ($R_c^2$). Post-hoc pairwise comparisons were then performed for each level of each factor, with Tukey's adjustment method. A similar ANOVA was performed on airborne OTU richness, including habitat (forest and vineyard), sampling date, sampling site, and their interactions.

Dissimilarities in composition between samples were represented by non-metric multidimensional scaling analysis (NMDS) and were analysed by permutational multivariate analyses of variance (PERMANOVA), including the same fixed factors as the ANOVAs, with sampling sites treated as strata. We dealt with complex interactions in PERMANOVA results by calculating post-hoc PERMANOVAs, including sampling date, sampling site and their interaction, separately for each host plant species (or habitat for airborne samples). We then corrected the P-values for multiple testing, as described by *Benjamini & Yekutieli (2001)*.

## RESULTS

### Taxonomic description of foliar and airborne fungal communities

In total, we obtained 7,946,646 high-quality sequences, which clustered into 4,360 OTUs. Overall, 867 OTUs, corresponding to 4,600,179 sequences (57.9% of the raw OTU table) were not taxonomically assigned to fungi by BLAST. Among them, 4,451,913 sequences were assigned to plant sequences (Tracheophyta division), principally *Vitis* (59%), and *Carpinus* (35%) genus, showing that fITS7-ITS4 primers are not specific of fungi. These OTUs were removed. The negative controls contained 29,857 fungal sequences clustering into 337 OTUs. There is no consensus on how to deal with OTUs found in negative controls (*Nguyen et al., 2015*; *Galan et al., 2016*). It is difficult to distinguish real contaminations (sequences originating from the people who performed the experiments, the laboratory environment and the DNA extraction kit) from cross-contaminations between samples, occurring during the DNA extraction, amplification and sequencing (*Esling, Lejzerowicz & Pawlowski, 2015*; *Galan et al., 2016*). It is highly probable that OTUs assigned to *Erysiphe alphitoides*, the agent responsible for the oak powdery mildew (1.5% of the negative control sequences; *Jakuschkin et al., 2016*) or *Botrytis cinerea*, responsible for the grey mold on grapes (1.2%; *Jaspers et al., 2016*) are likely cross-contaminations because they are strongly related to a specific host. Moreover, the removal of very abundant OTUs strongly altered the taxonomic composition of the samples, and removed some species known to be abundant on leaves such as *Aureobasidium pullulans*, known as very abundant on grapevine (*Pinto & Gomes, 2016*). We thus decided

to retain all these OTUs in the dataset. Two samples containing very few sequences (< 300 sequences) were removed. These samples corresponded to grapevine leaves collected at the Couhins site, in May. The first was collected in the centre of the vineyard, and the other was collected at its edge. Finally, the OTU table used for the analyses contained 196 samples and 3,487 fungal OTUs, corresponding to 3,316,156 sequences. The number of sequences per sample ranged from 424 to 96,276, with a mean of 16,919. This OTU table was used for taxonomical description. Richness, Bray-Curtis and Jaccard averaged indices were calculated over 100 rarefactions of this OTU table, at a threshold of 420 sequences per sample.

The fungal communities of bioaerosols and leaves from forest trees and grapevines were dominated by ascomycetes (Fig. 2). The sequences assigned to Ascomycota division accounted for 85.7% of all the sequences, followed by Basidiomycota division (11.3%). Overall, 3.0% of the total sequences remained unassigned at the division level. Airborne and foliar samples shared 1,440 OTUs (Fig. 3), but there was a significant difference in the composition of foliar and airborne fungal communities (PERMANOVA F = 20.15, P = 0.001). The 10 most abundant fungal OTUs were shared by airborne, forest foliar and grapevine foliar communities, but their relative abundance differed between each compartment (Table 1).

## Variations in the richness of foliar and airborne fungal communities at the landscape scale

ANOVA revealed a significant effect of the interaction between host plant species and sampling date on the richness of foliar fungal communities (Table 2). Differences in fungal community richness between plant species were not significant in May and July (Figs. 4 and S1). In October, grapevine stocks had significantly less rich foliar fungal communities than oak (post-hoc tests: P < 0.0001; Fig. 4) and hornbeam trees (P < 0.0001), but the richness of their fungal communities did not differ significantly from that of chestnut trees (P = 0.147; Fig. S1). Hornbeam leaves harboured the richest communities of all the plant species considered (post-hoc tests: P < 0.0001 between hornbeam and chestnut, P = 0.0003 between hornbeam and oak, P < 0.0001 between hornbeam and grapevine; Fig. S1).

ANOVA post-hoc tests also revealed a significant decrease in fungal species richness in grapevine over the course of the growing season (P < 0.0001 for each pairwise comparison; Fig. 4). Seasonal variations in fungal richness were less marked in oak (P = 0.081, P = 0.999 and P = 0.004, respectively between May and July, July and October, May and October), chestnut (P = 0.011, P = 0.997 and P = 0.0002, respectively) and hornbeam (P = 1.00, P = 0.144 and P = 0.185, respectively).

ANOVA also revealed a significant effect of the interaction between host plant species and edge on the richness of foliar fungal communities (Table 2). The richness of foliar fungal communities was significantly higher at the edge in oak (P = 0.002), but not in hornbeam (P = 0.100), chestnut (P = 0.139), or grapevine (P = 0.790) (Fig. S2).

Habitat had a significant effect on the richness of airborne fungal communities (Table 2), which was significantly higher in forests than in vineyards.

Conclusions were similar on models performed without rarefaction (Table S4).

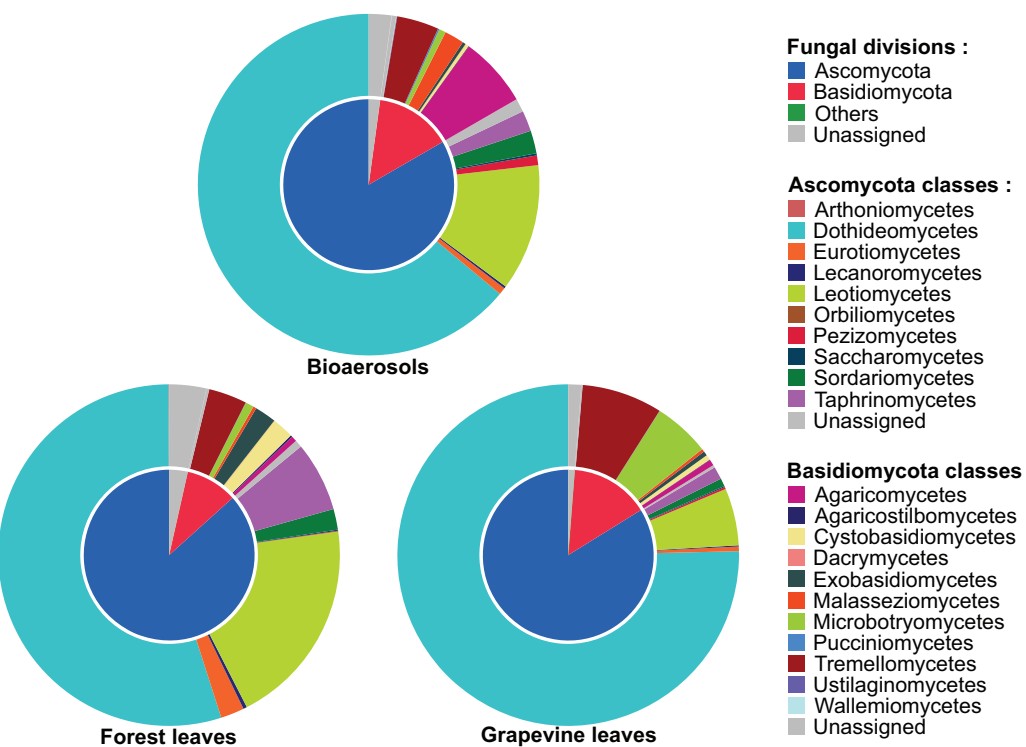

**Figure 2 Taxonomic composition of the airborne and foliar fungal communities in forest and vineyard habitats.** The inner disc shows the proportion of sequences assigned to each taxonomic division, and the outer disc the proportion of sequences assigned to each class of the Ascomycota and Basidiomycota divisions.

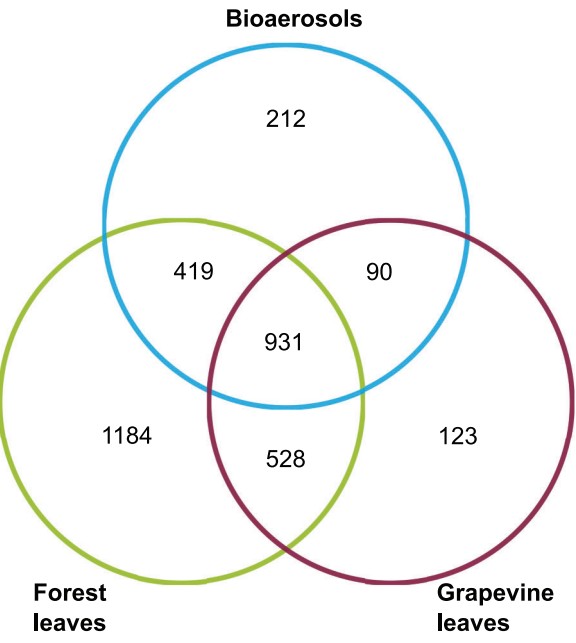

**Figure 3 Venn diagram giving the number of OTUs shared between the airborne, forest foliar and vineyard foliar communities.**

**Table 1 Taxonomic assignment of the 10 most abundant OTUs by the online BLAST analysis against the GenBank database.** The environmental and metagenome sequences were excluded. Identity is the percentage identity between the OTU representative sequence and the closest matching sequence in GenBank. Taxa shown as unassigned at the species level (*sp.*) indicate OTUs assigned to at least two species of the same genus with identical e-value. Relative abundance are percentage of abundance of each data subset and brackets contain the rank of the OTU in each data subset.

| Closest match | | | Relative abundance in percent (rank) | | | |
|---|---|---|---|---|---|---|
| GI number | Identity | Putative taxon | Total | Airborne | Forest leaves | Grapevine leaves |
| 1034220623 | 100 | *Aureobasidium pullulans* | 15.48 | 3.8 (**4**) | 12.6 (**1**) | 55.9 (**1**) |
| 1031917897 | 100 | *Cladosporium sp.* | 8.01 | 29.8 (**1**) | 2.7 (**11**) | 2.4 (**5**) |
| 1049480240 | 85.6 | *Collophora hispanica* | 5.64 | 1.7 (**7**) | 7.4 (**2**) | 1.1 (**13**) |
| 61619908 | 100 | *Ramularia endophylla* | 4.72 | 0.6 (**20**) | 6.4 (**3**) | 1.4 (**12**) |
| 1035371449 | 100 | *Cladosporium sp.* | 4.51 | 13.7 (**2**) | 2.3 (**13**) | 1.8 (**7**) |
| 530746702 | 100 | *Stromatoseptoria castaneicola* | 3.48 | 0.3 (**31**) | 4.8 (**4**) | 0.9 (**15**) |
| 626419142 | 99.5 | *Taphrina carpini* | 3.35 | 1.3 (**9**) | 4.3 (**6**) | 0.7 (**19**) |
| 1024249962 | 100 | *Erysiphe sp.* | 3.17 | 0.3 (**33**) | 4.4 (**5**) | 0.8 (**16**) |
| 61619940 | 100 | *Naevala minutissima* | 2.99 | 1.2 (**10**) | 3.8 (**8**) | 0.7 (**20**) |
| 961502090 | 91.0 | *Zeloasperisporium searsiae* | 2.93 | 0.2 (**46**) | 4.1 (**7**) | 0.6 (**21**) |

**Table 2 Effect of sampling date (May, July or October), host species (oak, hornbeam, chestnut or grapevine) or habitat (vineyard or forest), edge (habitat centre or center) and their interaction on OTU richness in foliar and airborne fungal communities, assessed using a type III ANOVA.** In both models, sampling site was included as a random variable. $R_m^2$ is the marginal coefficient of determination (for fixed effects) and $R_c^2$ the conditional coefficient of determination (for fixed and random effects).

| | F | P-value | $R_m^2$ ($R_c^2$) |
|---|---|---|---|
| | Foliar OTU richness | | |
| Date | 44.49 | **< 0.001** | 0.64 (0.71) |
| Species | 14.97 | **< 0.001** | |
| Edge | 17.21 | **< 0.001** | |
| D × S | 23.42 | **< 0.001** | |
| D × E | 0.11 | 0.894 | |
| S × E | 6.72 | **< 0.001** | |
| D × S × E | 1.13 | 0.347 | |
| | Airborne OTU richness | | |
| Date | 1.07 | 0.362 | 0.34 (0.52) |
| Habitat | 10.19 | **0.004** | |
| Edge | 4.20 | 0.052 | |
| D × H | 0.86 | 0.436 | |
| D × E | 1.40 | 0.267 | |
| H × E | 0.01 | 0.912 | |
| D × H × E | 1.678 | 0.209 | |

**Note:**
Bold values are the significant ones.

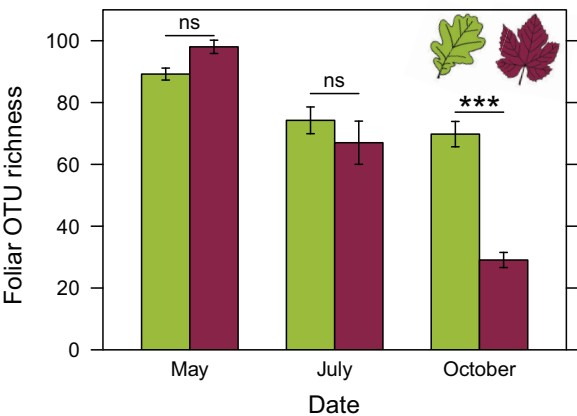

**Figure 4 Richness of foliar fungal community in oak (green) and grapevine (red), depending on the sampling date.** Error bars represent the standard error.

## Variations in the composition of foliar and airborne fungal communities at the landscape scale

PERMANOVA revealed a significant effect of the interaction between host plant species and sampling date on the composition of foliar fungal communities (Table 3). Bray-Curtis dissimilarities between oak and grapevine foliar fungal communities increased over the course of the growing season (mean ± SD; 0.47 ± 0.07 in May, 0.67 ± 0.09 in July and 0.91 ± 0.06 in October). These results are illustrated by NMDS (Fig. 5A). Bray-Curtis dissimilarities also increased between each pair of host species (Table S2; Fig. S3A). Similar results were obtained with the Jaccard dissimilarity index (Table S3; Fig. S3B).

PERMANOVA also revealed significant edge effects on the composition of foliar fungal communities, in interaction with host plant species and sampling date. Post-hoc PERMANOVAs computed separately for each host species indicated differences in community composition between the edge and centre of the forest for oak and hornbeam, in interaction with sampling date (F = 1.68, P = 0.031 and F = 1.85, P = 0.044, respectively). The composition of the fungal community did not differ between the edge and the centre of the habitat for chestnut (F = 2.27, P = 0.25) or grapevine (F = 0.92, P = 1). Finally, PERMANOVA analysis of Bray-Curtis dissimilarities revealed a significant effect of sampling date on bioaerosol composition (Table 3; Fig. 5B). Similar results were obtained for Jaccard dissimilarity (Table S3). Overall, similar results were also obtained without rarefying (Table S5).

## DISCUSSION

To our knowledge, this is the first time that the spatial structure and the temporal dynamic of foliar and airborne fungal communities are assessed simultaneously at the landscape scale. We studied a landscape mosaic consisting of two main habitats, vineyard and forest patches. We expected that repeated dispersal events between habitat patches would homogenize the foliar communities over the course of the growing season. We expected the homogenization to be greater along habitat edges, where grapevine stocks and forest trees are closer to each other.

**Table 3 Effect of sampling date (May, July or October), host species (oak, hornbeam, chestnut or grapevine) or habitat (vineyard or forest), edge (habitat centre or center) and their interaction on the composition of foliar and airborne fungal communities, assessed using a PERMANOVA.** In both models, sampling site was included as a stratification variable.

| | F | $R^2$ | P-value |
|---|---|---|---|
| | Foliar fungal community composition | | |
| Date | 10.13 | 0.078 | **0.001** |
| Species | 13.70 | 0.158 | **0.001** |
| Edge | 3.94 | 0.015 | **0.001** |
| D × Sp | 6.92 | 0.160 | **0.001** |
| D × E | 2.05 | 0.016 | **0.007** |
| Sp × E | 2.22 | 0.026 | **0.001** |
| D × Sp × E | 1.08 | 0.025 | 0.239 |
| | Airborne fungal community composition | | |
| Date | 2.94 | 0.157 | **0.001** |
| Habitat | 1.54 | 0.041 | 0.062 |
| Edge | 0.68 | 0.018 | 0.827 |
| D × H | 0.95 | 0.051 | 0.418 |
| D × E | 0.66 | 0.035 | 0.938 |
| H × E | 0.77 | 0.020 | 0.684 |
| D × H × E | 0.71 | 0.038 | 0.878 |

**Note:**
Bold values are the significant ones.

Accordingly, we found that 26% of the OTUs are shared between airborne and foliar fungal communities. The most abundant ones are principally generalist species, such as *Aureobasidium pullulans*, *Cladosporium sp.* or *Eppicoccum nigrum*, which were already found as abundant in the microbiome of many species (*Jumpponen & Jones, 2009*; *Zambell & White, 2015*; *Pinto & Gomes, 2016*). This result confirms that many fungal species disperse through the atmosphere (*Lindemann et al., 1982*; *Brown & Hovmøller, 2002*; *Bulgarelli et al., 2013*). Moreover, while the richness of airborne fungal communities was higher in forest patches than in adjacent vineyards, their composition did not differ significantly, whatever the season. This lack of spatial variation in airborne fungal communities could account for the high similarity between foliar fungal communities of grapevine and forest tree species at the beginning of the growing season. Flushing leaves in May receive similar pools of fungal species through airborne dispersal, whatever the habitat and the host plant species. Our results suggest that dispersal of foliar fungal communities is not limited at the landscape scale. Similar patterns were already observed at far larger spatial scales. The atmosphere is indeed considered as a continental and inter-continental corridor for the dispersal of microorganisms (*Finlay, 2002*; *Brown & Hovmøller, 2002*; *Womack, Bohannan & Green, 2010*; *Barberán et al., 2014*), resulting in global patterns across continents. However, our results contrast with the strong dispersal limitation observed at smaller scale (*Bowers et al., 2013*). *Peay, Garbelotto & Bruns (2010)* found that ectomycorhizal richness is lower in small tree patches located 1 km away from large tree patches than nearer ones. *Dickie & Reich (2005)* showed that the

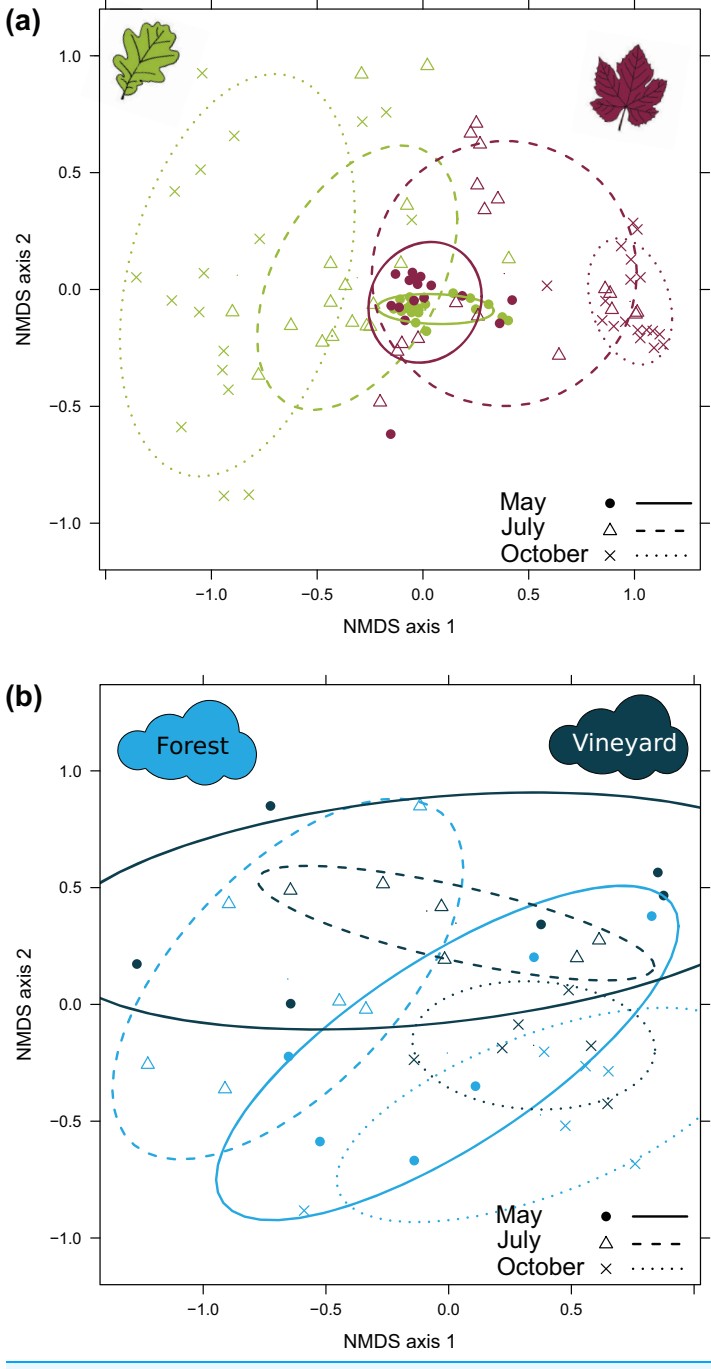

**Figure 5 NMDS representing dissimilarities in the composition of fungal communities.** (A) Dissimilarities in the composition of foliar fungal communities between the host species (oak in green and grapevine in red), depending on the sampling date. The other two forest species are not shown here, to make the figure easier to read, and are presented in Fig. S2. The stress value associated with this representation was 0.170. (B) Airborne fungal communities between the habitat (forest in light-blue and vineyard in dark-blue), depending on the sampling date. The stress value associated with this representation was 0.188. Dissimilarities between samples were computed with the Bray-Curtis index, averaged over 100 random rarefactions of the OTU table. The confidence ellipsoid at the 0.68 level is shown, for all combinations of these two factors.

abundance and richness of ectomycorhizal fungi decreased up to 20 m away from the forest edge. *Galante, Horton & Swaney (2011)* also showed that 95% of ectomycorhizal spores fell within 58 cm from the source. While the dispersal of ectomycorhizal fungi can differ from the foliar fungi because of differences in the height of spore emission (*Schmale & Ross, 2015*), our failure to detect such dispersal limitation at small spatial scales can be explained by the short time of sampling of airborne communities (30 min), which can be insufficient to properly characterize the airborne fungal composition of the whole season.

Against expectation, we found that the composition of the foliar fungal communities of forest tree species and grapevine increasingly diverged from May to October. Besides, a severe decline in the richness of foliar fungal communities was observed in grapevine over the course of the growing season, but not in forest tree species. Despite an identical pool of airborne fungi in vineyards and forests, the selective pressures exerted on foliar fungal communities therefore differ between both habitats. These selective pressures can be exerted by several factors, including the host species, the microclimate and the agricultural practices. Host-specificity has been demonstrated in foliar fungal communities (*Kembel & Mueller, 2014*; *Lambais, Lucheta & Crowley, 2014*; *Meiser, Bálint & Schmitt, 2014*). Our results paralleled these findings: in forest patches, foliar fungal communities significantly differ among tree species at the end of the growing season. Seasonal variations in leaf physiology could also account for the observed temporal variations in foliar communities, especially the richness decline in grapevine fungal foliar communities. Older grapevine leaves indeed produce larger amounts of phytoalexins and tend to be more resistant to pathogens (*Steimetz et al., 2012*). Selection by the habitat can also be exerted by the microclimate (*Vacher et al., 2016*). Harsher microclimatic conditions in vineyards than in forests, especially in the summer, could account for the decline in fungal species richness in vineyards but not in forests. Particularly, greater exposure to UV and higher air temperatures in vineyards could decrease the survival of foliar fungi on grapevine leaves. By contrast, tree cover provides a milder microclimate which could be more suitable to foliar micro-organisms. Finally, selection by the habitat can be exerted by agricultural practices. A few studies showed that fungicide applications can reduce the diversity and alter the composition of the foliar microbial community (*Gu et al., 2010*; *Moulas et al., 2013*; *Cordero-Bueso, Arroyo & Valero, 2014*; *Karlsson et al., 2014*). However, several other studies showed that the foliar fungal communities of grapevine are highly resilient to some chemical or biological pesticides (*Walter et al., 2007*; *Perazzolli et al., 2014*; *Ottesen et al., 2015*). Further research is required to assess the influence of fungicide applications on the observed decline in the richness of foliar fungal communities.

Our study also showed, for the first time, significant edge effects on foliar fungal community assemblages. A higher level of foliar fungal community richness was found in oak trees growing at the edge of the forest than in oak trees growing 100 m away. Significant differences in community composition between the edge and the centre of the forest were also found for oak and hornbeam. Variations in microclimate and leaf physiology along the forest edge (*Chen, Franklin & Spies, 1993*; *Zheng et al., 2005*; *Kunert et al., 2015*) are more likely to account for this result than species dispersal

from vineyards to forest patches, since the foliar fungal communities of the two habitats diverged over the course of the growing season. The absence of edge effect in grapevine foliar fungal communities suggests that dispersal of fungal species from forests to vineyards has little influence on community composition and richness. This result contrasts with the findings of many studies on macro-organisms, reporting that dispersal events between managed and non-managed habitats shape communities and influence ecosystem functioning and services (*Thomson & Hoffmann, 2009*; *Rusch et al., 2010*; *Thomson et al., 2010*; *Chaplin-Kramer et al., 2011*; *Blitzer et al., 2012*).

## CONCLUSIONS

Our results suggest that dispersal events between habitat patches are not major drivers of foliar fungal communities at the landscape scale. Selective pressures exerted in each habitat by the plant host, the microclimate and the agricultural practices play a greater role, and might account for the differentiation of foliar fungal communities between habitats. However, our experimental design does not allow us to assess the relative influence of each factor in shaping foliar fungal communities. Our results suggest that the leaves of broad-leaf species are colonised by similar pools of airborne micro-organisms at the beginning of the growing season. The composition of foliar fungal communities then diverges between habitat patches and between plant species within the same habitat. In contrast, airborne communities remain similar between habitats. Overall, our results support those of *Redford et al. (2010)* and *Morrison-Whittle & Goddard (2015)* which indicated that selection predominates over dispersal in structuring plant microbial communities.

## ACKNOWLEDGEMENTS

We thank the vineyard managers for allowing us to carrying out these experiments in their domains: Nicolas Lesaint in Château Reignac, Mathieu Arroyo in Château Couhins, and Antoine Le Grix de la Salle in Château Grand-Verdus. We thank Morgane Petitgenet for assistance with sample collection, and Olivier Fabreguettes and Jessica Vallance for providing help and advice for DNA extraction and bioinformatic analyses. We thank the entire LGC genomics team for assistance and advice concerning sequencing.

### Funding

This study was carried out with financial support from the French National Research Agency (ANR) in the frame of the Investments for the future Programme, within the COTE Cluster of Excellence (ANR-10-LABX-45). The funders had no role in study design, data collection and analysis, decision to publish, or preparation of the manuscript.

### Grant Disclosures

The following grant information was disclosed by the authors:
French National Research Agency (ANR).
COTE Cluster of Excellence: ANR-10-LABX-45.

## Competing Interests

The authors declare that they have no competing interests.

## Author Contributions

- Thomas Fort performed the experiments, analyzed the data, wrote the paper, prepared figures and/or tables, reviewed drafts of the paper.
- Cécile Robin conceived and designed the experiments, performed the experiments, wrote the paper, reviewed drafts of the paper.
- Xavier Capdevielle performed the experiments.
- Laurent Delière conceived and designed the experiments, performed the experiments, reviewed drafts of the paper.
- Corinne Vacher conceived and designed the experiments, performed the experiments, wrote the paper, reviewed drafts of the paper.

## DNA Deposition

The following information was supplied regarding the deposition of DNA sequences:

European Nucletide Archive dataset (PRJEB13880): https://www.ebi.ac.uk/ena/data/view/PRJEB13880.

## Data Deposition

The raw data is available as Supplemental Dataset Files.

## Supplemental Information

Supplemental information for this article can be found online at http://dx.doi.org/10.7717/peerj.2656#supplemental-information.

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
