# Peer review of "Foliar fungal communities strongly differ between habitat patches in a landscape mosaic"

_PeerJ, doi:10.7717/peerj.2656_

## Round 0.1 · original submission · Major Revisions

I agree with the two reviewers that the manuscript is well-written and presented. However, I share the concern of Reviewer 1 regarding the impact of rarefaction of the data. I urge you to consider Reviewer 1's suggestion of incorporating read number into the analysis, or to otherwise examine the impact of rarefaction on your conclusions. Reviewer 2 has also made important points about the fungicide treatment of the grapevines and concluding paragraph that must be addressed in a revised manuscript. Because both reviewers raise questions about the analyses, my editorial decision is Major Revisions.

I have attached an annotated pdf with minor corrections and my questions about the rarefaction.

·

Basic reporting

The article is written in clear English.
The introduction is sufficient to place the work into context.
I find the figures relevant.
I consider the work is 'self-contained'.
I very much like the nice and easy figures.

Experimental design

The work is original primary research.

There are clear hypotheses listed and they are addressed with appropriate methodology.

The investigation seems rigorous and it meets high technical standards in my opinion. I particularly liked that they sterilized their sampling tools.

I do not agree that the authors use rarefaction to equalize read numbers in samples. Read number differences can be easily treated by including them into the analyses, and the modeling framework used here is perfectly suitable for this.

I very much welcome that the authors made their analysis script public. This should be the standard practice of every research article.
I encourage the authors to provide a schematic drawing of their experimental design - this will help understanding a lot.

Validity of the findings

I find the data mostly robust and controlled, but I have concerns how the authors treated the OTUs that showed up in the negative controls. It is nothing unusual that OTUs sometimes show up in the negatives, but I don't understand the why authors decided to keep these potentially contaminant OTUs in the analyses. I would normally expect that they are completely removed from the entire dataset. Additionally, the OTUs that show up in many samples including the negative controls are later discussed as generalists - which is fine, but they may also come from a contamination event that influenced all samples.

Since most of the results and their discussions are concerned with differences among samples/habitats/seasons I think a potential contamination with some common OTUs should not influence the results and conclusions considerably.

Line 207 - a p=0.001 seems as indication of statistical significance.

·

Basic reporting

No issues with basic reporting; please see specific comments below.

Experimental design

Some potential issues with experimental design; please see specific comments below.

Validity of the findings

Some potential issues with validity of the findings; please see specific comments below.

Additional comments

Overall, I thought that this was a very well written and well organized manuscript that asks an interesting question about fungal community similarities and (by inference) dispersal patterns in patchy landscapes. The spatial design of the field sampling approach seemed generally appropriate and well thought out. I appreciated the fact that the authors sampled not only leaves but also airborne propagules. The sequencing, bioinformatic, and statistical methods used are up to date and appropriate, and the raw data files in addition to the R scripts used for analysis are included with the manuscript as supplemental files. The figures are well-designed and visually appealing. I enjoyed reading the paper and look forward to seeing it published.

My major concern with the study has to do with the fungicide treatments applied to the grape plants during the growing season and the effect of these differential treatments across sites. I recognize that the authors may not have had the option to study vineyards that did not apply fungicides (do such things exist? Even organic vineyards apply sulfur, as far as I understand it). However, I think that a more careful effort needs to be made dealing with this potentially confounding effect on the findings and their interpretation. For example, are the successive applications of fungicide the reason for the decrease in diversity over the growing season? It seems difficult with the current study design to decouple effects of aging leaves or foliar community succession patterns from chemical treatment effects. This confounding effect could potentially (based on the assumption that fungicide applied to the fields did not drift into the forest) be excluded by focusing exclusively on the forest trees sampled, since they did not receive these chemical treatments.

The authors are admirably open about this issue and do include information in a supplementary table about what was applied to each field and when, and they devote a paragraph in the discussion to this issue, but I didn't get the sense that the potentially confounding nature of these treatments was adequately handled in the text or in the analyses themselves.

The other major comment I had is about the conclusion. It seems to me that the authors' claims are a bit too broad about 'everything being everywhere' given their data. The rest of the manuscript is very well written, appropriately speculative, and well-cited, but I think the discussion/conclusion in particular would benefit from a slightly more careful/nuanced discussion of dispersal and dispersal limitation and their role in structuring phyllosphere or other kinds of fungal communities. In my opinion, scale of dispersal seems like a key issue when considering dispersal limitation in fungal communities, and my sense is that this is not directly addressed by the current manuscript at scales larger than individual sites. Relatedly, I think it would be nice to see some discussion of the series of papers by Peay et al. on dispersal limitation on fungal communities. I realize that the current manuscript is focused on phyllosphere fungi, and that the Peay et al. studies are primarily focused on mycorrhizal fungal communities, but I think the references are nonetheless relevant and worth consideration in the discussion/conclusion. I think that the authors come to a reasonable conclusion given their results, but perhaps they present an over-simplified picture of landscape dispersal. Certainly the last sentence of the conclusion seems a bit too much of a reach given that the scale-dependence of patterns across the landscape (especially at larger spatial scales) could be more important than the manuscript implies.

Table 1. Caption typo: OUT should be OTU. I'd add the (SRA?) Project Accession Number to the main text - currently it is only in this table legend (unless I missed it).

Figure 2. I wonder how the fungal OTU that are not found in bioaerosols but are found in grapevine leaves colonize those leaves. Could it be just that the aerosol sampling didn't capture particular temporal pulses of particular fungal species since they are effectively point measurements (vs integrated samples over long periods of time)? Or...?

Citations mentioned:
Peay, K. G., T. D. Bruns, P. G. Kennedy, S. E. Bergemann, and M. Garbelotto. 2007. A strong species-area relationship for eukaryotic soil microbes: island size matters for ectomycorrhizal fungi. Ecology Letters 10:470-480.

Peay, K. G., M. Garbelotto, and T. D. Bruns. 2010. Evidence of dispersal limitation in soil microorganisms: Isolation reduces species richness on mycorrhizal tree islands. Ecology 91:3631-3640.

Peay, K. G., M. I. Bidartondo, and A. E. Arnold. 2010. Not every fungus is everywhere: scaling to the biogeography of fungal--plant interactions across roots, shoots and ecosystems. New Phytologist 85:878-882.

---

## Round 0.2 · accepted · Accept

The reviewers pointed out minor grammatical errors in the revised discussion. I have made suggested corrections in the attached pdf.

·

Basic reporting

I had no issues with the previous version of the MS in this respect.

Experimental design

I welcome that the authors included a figure of the experimental design - it makes understanding so much easier! It is also great that they performed additional analyses that accounts for reads in the modelling framework - the results of these supported their earlier conclusions.

Validity of the findings

I think that the additional information the authors include about the treatment of negatives is a reasonable argument for their decisions.

Additional comments

In general I feel that the authors provided adequate answers to my comments. The changes improved the manuscript.

typo - line 355: can be explain by => can be explained by

·

Basic reporting

There are a few minor grammatical errors that should be fixed before publication. The figures are very nicely done. Overall, the paper is well written and organized.

Experimental design

My only remaining concern is with the way the sequences in the negative controls were handled, but I also think that the authors have made a reasonable decision given their data, and have added text discussing this choice, so perhaps it is ok as is.

Validity of the findings

Much improved in terms of the scope of the results and the addition of appropriate caveats.

Additional comments

Much improved, thanks for making the changes. Very much appreciate the inclusion of the supplemental Rmd script.